# In Vitro Methods to Decipher the Structure of Viral RNA Genomes

**DOI:** 10.3390/ph14111192

**Published:** 2021-11-20

**Authors:** Cristina Romero-López, Sara Esther Ramos-Lorente, Alfredo Berzal-Herranz

**Affiliations:** Instituto de Parasitología y Biomedicina López-Neyra (IPBLN-CSIC), Av. del Conocimiento 17, 18016 Armilla, Granada, Spain; seramos@ipb.csic.es

**Keywords:** RNA structure, interactome, RNA probing, long-distant RNA-RNA interactions, molecular interference, SHAPE

## Abstract

RNA viruses encode essential information in their genomes as conserved structural elements that are involved in efficient viral protein synthesis, replication, and encapsidation. These elements can also establish complex networks of RNA-RNA interactions, the so-called RNA interactome, to shape the viral genome and control different events during intracellular infection. In recent years, targeting these conserved structural elements has become a promising strategy for the development of new antiviral tools due to their sequence and structural conservation. In this context, RNA-based specific therapeutic strategies, such as the use of siRNAs have been extensively pursued to target the genome of different viruses. Importantly, siRNA-mediated targeting is not a straightforward approach and its efficiency is highly dependent on the structure of the target region. Therefore, the knowledge of the viral structure is critical for the identification of potentially good target sites. Here, we describe detailed protocols used in our laboratory for the in vitro study of the structure of viral RNA genomes. These protocols include DMS (dimethylsulfate) probing, SHAPE (selective 2′-hydroxyl acylation analyzed by primer extension) analysis, and HMX (2′-hydroxyl molecular interference). These methodologies involve the use of high-throughput analysis techniques that provide extensive information about the 3D folding of the RNA under study and the structural tuning derived from the interactome activity. They are therefore a good tool for the development of new RNA-based antiviral compounds.

## 1. Introduction

Viral RNA genomes are dynamic entities. During the infection, they must be translated, copied, and packaged to produce a new generation of virions. Maintaining a proper balance between all these processes involves precise regulatory mechanisms that require the interaction of different genome-encoded elements, giving rise to the so-called interactome [1,2]. These genome-encoded elements fold autonomously and show high structural conservation among the different viral isolates [3].

During viral replication, a wide spectrum of mutants is generated due to the high error rates of the viral RNA-dependent RNA polymerases. This phenomenon hinders the development of effective antiviral drugs with sustained virological responses. The fact that RNA folding and the RNA interactome in viral genomes are well conserved makes it tempting to propose them as potential targets for the development of new antiviral treatments [4].

RNA-based therapies are promising antiviral strategies. Particularly, the siRNA technology, which has been extensively tested against a wide variety of viruses (for a review see [5]). These studies have proved that the efficiency of different siRNAs against the same target gene can be different; confirming that the random design of siRNAs is not a valid strategy when it comes to obtaining efficient inhibitors. One of the critical rules to take into account is related to the folding and accessibility of the target region [6,7,8,9,10]. Many of the highly conserved regions in viral genomes, which show a compact folding, can be refractory to siRNA targeting. For that reason, the biochemical and biophysical study of RNA folding, together with the use of bioinformatics strategies must be accomplished to design efficient RNA-based inhibitors. This will provide an excellent starting point for the development of antiviral strategies, which is a real need for many RNA viruses that are responsible for major human diseases today. Besides the siRNA-based technology, other RNA-based therapies are promising candidates that take advantage of the study of RNA folding. These include the use of aptamers, antisense oligonucleotides, or ribozymes.

This article describes in detail protocols focused on the application of high-throughput probing methods followed in our laboratory to decipher RNA-RNA interactions in viral genomes.

## 2. RNA Probing

RNA probing focuses on the determination of certain structural features of the target RNA molecule, such as Watson-Crick base pairing, sugar-phosphate backbone flexibility, or solvent accessibility. These properties can be studied independently by using different reagents. Thus, the greater the number of features analyzed, the greater the accuracy of the final structural model [11,12]. The specificities of the most commonly used chemical probing reagents, both in vitro and in vivo, are listed in Table 1 and Table 2.

Most chemical probing reagents show easy access to the reactive groups of RNA due to their small size and can attack a wide range of positions (Figure 1). They also exhibit high diversity in their reaction timescale. Some of them can react at low timescales, identifying residues with slow electronic dynamics capable of stabilizing the RNA architecture. In contrast, other reagents target “one-sided” stacked nucleotides with fast electronic dynamics, a typical conformation seen in bulges, turns, closing helix pairs, and long-range stacking interactions [13,14,15,16] (Table 2 and Figure 1). Combining the reactivity data derived from different probing reagents renders a complete fingerprint map, which summarizes the non-canonical and stacking interactions that define the three-dimensional architecture of the RNA molecule.

In vitro, we have applied different probing strategies to analyze subgenomic HCV RNA constructs (Figure 2). DMS treatment and SHAPE assays with different timescale reacting reagents have provided remarkable and reproducible data [17,18,19]. Experimental details of the dimethyl sulfate (DMS) and N-methyl isatoic anhydride (NMIA) probing assays are described below.

### 2.1. Basic Protocol 1: RNA Probing with DMS

Probing RNA with DMS provides information from Watson–Crick and Hoogsteen pairs. It can be used over a broad pH range with minor changes in reactivity, making it a suitable tool for RNA probing under different experimental conditions, including intracellular environments [20].

DMS modifies unpaired A, C, and G residues by introducing methyl groups at positions N1, N3, and N7, respectively [21]. Methylated residues are detected by primer extension reaction (see Section 3). A prior aniline-induced strand scission is required to identify modified G residues [22] (Figure 2A and Figure 3).

1. Denature 1 pmol of purified RNA per reaction by heating at 95 °C for 2 min.

2. Transfer the sample to an ice/water bath and incubate for 15 min.

3. Distribute 1 pmol aliquot of denatured RNA into new tubes. It should be noted that at least two samples must be prepared to be assayed in the absence (−) or presence (+) of DMS. This is required to compare both reverse transcription (RT) patterns.

4. Add folding buffer and proceed to renature the RNA molecules by incubating at the desired temperature for 5 min. 37 °C is usually a good option, although other conditions can be further tested.

5. Add 1 µg of tRNA to each reaction tube to avoid extensive RNA modification by DMS.

6. Initiate the probing reaction by adding 1–5 µL of freshly diluted DMS in ethanol (1:5) [(+) DMS reaction] or net ethanol [(-) DMS reaction], and mix by gentle pipetting. In this step, a final reaction volume of 15–20 µL is recommended. Incubate the reactions at 37 °C during 60–90 s. The concentration of the probing reagent should be optimized for each RNA problem. To assay different concentrations of freshly prepared probing reagent, starting with the indicated concentration might be necessary. One to three modified nucleotides per molecule is desirable. A low concentration of the probing reagent results in incomplete probing of the (+)RNA sample, so the probe concentration should be increased. Conversely, an excess of probing reagent may result in the absence of full-length products and a very low signal for distant nucleotides. In this case, reducing the concentration of the chemical reagent (around two-fold) may solve the problem.

7. Complete up to 150 µL with sterile RNase-free distilled water and stop DMS-mediated RNA modification by the addition of 0.1 volumes of 3 M sodium acetate, pH 5.2.

8. Proceed to RNA precipitation by the addition of three volumes of cold (−20 °C) absolute ethanol and incubate the samples at −80 °C for 30 min or at −20 °C overnight. In this step, an inert carrier such as glycogen can be supplemented to improve RNA precipitation.

9. Centrifuge RNA samples during 30 min at 12,000× *g* at 4 °C.

10. Carefully discard the supernatant and wash the pellet with 250 µL of 80% ethanol. Disruption of the fragile pellet can be avoided by omitting gentle vortexing or pipetting.

11. Centrifuge at 12,000× *g*, at 4 °C, for 10 min.

12. Discard the supernatant and repeat washing steps 9–11.

13. Discard the ethanol supernatant and briefly vacuum dry the RNA pellet to eliminate ethanol traces.

14. Use sterile RNase-free distilled water to dissolve the samples. The final volume must not exceed 10 µL.

### 2.2. Basic Protocol 2: RNA Probing with SHAPE Chemistry

In addition to the base-pairing pattern that can be inferred from the RNA probing analysis with DMS, it may also be desirable to obtain data about conformationally dynamic residues [23]. These residues may remain inaccessible to the solvent, but they are key players in determining the overall shape of the RNA molecule. The so-called SHAPE (selective 2′-hydroxyl acylation and primer extension) chemistry (Figure 2A) is the preferred strategy to reveal these residues. The SHAPE technique was described in 2005 by Merino et al. [24] and was proposed as an alternative strategy to traditional probing methods. Probing with classical reagents typically renders information of a sparse subset of nucleotides, while SHAPE chemistry takes advantage of the 2′OH group reactivity in the ribose moiety to get acylated nucleotides. Therefore, it can map any position in a target RNA.

SHAPE reagents (Table 2) can be used to probe the local flexibility of the RNA backbone in a nucleobase-independent manner [25] by the formation of covalent adducts with the 2′-OH group. Up to seven different SHAPE chemical probes have been reported, with variations in their half-lives ranging from 0.25 s to 73 min [14,15,23,26] (Table 2). Using two or more reagents will provide complementary structural information about the RNA molecule [12]. The present protocol describes the laboratory routine for the widely used NMIA SHAPE reagent [17,18,19]. The main steps resemble those described above for probing with DMS. However, the treatment conditions for NMIA are different.

1. Denature 1 pmol of the target RNA per reaction, as described above (see 2.1, step 1).

2. Proceed to RNA refolding by incubating it with folding buffer (100 mM Hepes/NaOH, pH 8.0; 100 mM NaCl; 5 mM MgCl_2_) for 5 min at the desired temperature. For timed SHAPE assays, in which conformational dynamics are studied, it may be required to test a wide range of temperatures. Note that NMIA reacts even at 95 °C [27].

3. Initiate NMIA-dependent acylation by adding 1 µL of freshly diluted NMIA reagent in DMSO. The optimum final concentration of NMIA may range from 2–100 nM, depending on the conformational features of the RNA molecule. In general, the ideal probe concentration renders less than one modified nucleotide per molecule. A concentration of 10 mM could be used as starting point, but it must be optimized for each RNA to be probed. Note that a non-treated sample, reaction (-), must be performed in the presence of 1 µL of net DMSO.

4. Incubate the reactions at the desired temperature for 5X NMIA half-life. The equation:half-life(min) = 360 × e [−0.102 × temperature(°C)]
can be used to get a close estimation of this parameter [28].

5. The formation of 2′-O-adducts is stopped by ethanol precipitation, as described (basic protocol 1 step 8).

6. The precipitated RNA must be washed twice with 80% ethanol (steps 9–11 from basic protocol 1).

7. Discard the supernatant and briefly dry the RNA pellet under vacuum conditions or allow it to air dry at room temperature for a few minutes to remove any ethanol trace amounts.

8. Resuspend the pellet in 10 µL of sterile RNase-free distilled water.

### 2.3. Basic Protocol 3: Chemical Interference (HMX)

The analysis of the secondary and tertiary RNA structure proposed in basic protocols 1 and 2 can be further complemented by introducing an additional chemical interference study with SHAPE reagents, as NMIA. This specific methodology is called HMX (2′-hydroxyl molecular interference) (Figure 3B). HMX is a powerful technique based on the random and sparse modification of the atomic positions in a given RNA at high temperatures (~95 °C) [27]. Some of the modifications introduced may disrupt the RNA folding, which can be monitored by different partitioning procedures, such as gel retardation, gel filtration, or affinity chromatography. After the purification of the different RNA conformers, nucleotide modification is detected by primer extension, as described for the DMS and SHAPE analysis (Figure 3B). Then, a comparison of the reverse transcription patterns for all conformers will allow for the detection of those residues that are essential for the acquisition of a given folding.

We have followed this strategy for analyzing quaternary structure in the HCV RNA genome [19]. A general working protocol can be outlined as follows:

1. RNA modification is accomplished under denaturing conditions by incubating 50 pmol of the construct with freshly diluted NMIA in DMSO for 3 min at 95 °C before cooling on ice for 2 min. The reaction proceeds in the presence of 100 mM HEPES pH 8.0 in a final volume of 20 µL. Note that a non-treated sample, (–) NMIA, must be prepared in net DMSO.

In this step, the optimal NMIA concentration must be determined experimentally. In our hands, an NMIA concentration of 20 mM is usually enough for probing ~5 µg of RNA.

2. Repeat the modification step twice, paying special attention to the replacement of the evaporated water to achieve the desired final volume.

3. The reactions are stopped by cooling on ice and subsequent RNA precipitation in the presence of 0.3 M of sodium acetate, pH 5.2, and three volumes of absolute ethanol.

4. Repeat steps 7–12 from Basic protocol 1.

5. Monitor RNA amount by UV spectrometry (A_260_).

6. Isolate different RNA conformers by applying electrophoretic mobility shift assays. Briefly, denature probed RNA molecules by heating at 95 °C for 2 min and subsequently cooling on ice for 15 min. Then, incubation at the optimal ionic and temperature conditions promotes the formation of different structural conformers. In our hands, incubating RNA samples at 37 °C for 30 min in a folding buffer works properly. Samples are loaded with non-denaturing loading buffer (0.04% xylene cyanole; 0.04%, bromophenol blue; 5% glycerol) on a non-denaturing polyacrylamide gel (5–10% acrylamide:bisacrylamide, 19:1, 1X TBM). The gel is run at 10 V/cm and 4 °C to avoid overheating.

7. RNA conformers are visualized using a UV transilluminator after GelRed^®^ staining or by UV shadowing. Bands are excised using a razor blade. It is noteworthy that the relative abundance of the different conformers should vary in the NMIA treated samples compared to the non-treated ones.

8. The gel slices are then soaked in elution buffer (1 mL:1 g; 0.5 M ammonium acetate; 0.1% sodium dodecyl sulfate, SDS;1 mM EDTA) and the RNA is eluted overnight at 4 °C by passive elution.

9. Purify the RNA by two consecutive phenol extractions and additional chloroform:isoamyl alcohol extraction.

10. Precipitate RNA samples as described in steps 7–12 from Basic protocol 1.

11. Use 10 µL of sterile RNase-free distilled water to resuspend the RNA pellet.

## 3. Identification of Modified Nucleotides

Modifications included in the RNA molecule during the probing reaction can be detected by end-labeling RNA or by primer extension-dependent procedures (Figure 3). In the first case, the RNA is labeled at either the 5′ or the 3′ end and subjected to those chemical modifications (Figure 3A) that are susceptible to specific cleavage. For example, DMS probing of guanine nucleotides requires aniline-mediated cleavage for further detection [29]. Then, separation of the RNA fragments is accomplished on high-resolution denaturing polyacrylamide gels. The use of appropriate size markers run in parallel allows the precise identification of the modified nucleotides.

Alternatively, a 5′ end-labeled oligonucleotide can be designed to hybridize to the target RNA and then be extended by RT reactions (Figure 3B) [23,24,28]. In this method, the modifications emerge as reverse transcription stop signals, generating a pool of cDNA fragments that are resolved by high-resolution or capillary electrophoresis [28,30].

The choice of one of these strategies depends on several items:(a)The RNA length is, likely the most determinant feature in choosing the primer extension method. It is useful for long RNAs because multiple primers can be used in separate reactions to analyze modifications along the whole RNA molecule.(b)Primer extension is also suitable for in vivo assays because it does not require previous RNA purification.(c)Finally, the primer extension reaction simplifies the protocol optimization since it can be used for detecting modifications performed by different reagents.

A main drawback of the primer extension-based strategy is that sequence-specific pausing of the reverse transcription reaction in the non-treated samples generates high background signals, which may be difficult for the analysis. Primer extension-based strategies also omit the information from those nucleotides within the oligonucleotide annealing site and generate intense noise signals caused by the shortest and the full-length cDNA products. These problems were successfully overcome with the strategy designed by Merino et al. [24]. These authors proposed the incorporation of specific, “structurally inert” cassettes to the target RNA during its synthesis. These cassettes include sequences that fold autonomously as stem-loop structures, to avoid interfering with the structure of the tested RNA [24]. The cassettes located at the 3′ and the 5′ ends displace the noise signal, which masks the putative specific signal produced by the residues at the ends of the RNA. The 3′ cassette also provides the primer binding site for initiation of the RT and prevents the interference with the fluorescent signal caused by the abundant non-annealed primer. We have found that the cassettes previously described by Merino et al., which were designed to map the tRNA^Asp^ molecules [24], are also adequate for the analysis of different HCV constructs carrying the IRES region, the CRE and/or the 3′UTR, as they do not interfere with the predicted folding [18,19]. The effect of these or any other cassette over the folding of the target RNA must be tested for each molecule under study.

Another issue to consider for the readout of probing experiments is the choice of the labeling agent. Currently, a wide variety of labels is available; however, ^32^P and fluorophore labeling are the most common options.

We have optimized the primer extension reactions with fluorescently labeled primers for HCV probing readout. Specifically, we have used the NED fluorophore for mapping both treated and untreated samples, while RNA sequencing reactions were performed with FAM or VIC-labelled oligonucleotides. This strategy allows to resolve of cDNA fragments by capillary electrophoresis and facilitates their quantification by the application of the QuShape software [31].

### 3.1. Primer Purification

1. Fluorescently labeled DNA primers used for reverse transcription assays must be purified on denaturing polyacrylamide gels. Briefly, add one volume of denaturing formamide loading buffer (47% deionized formamide; 0.012% xylene cyanole; 0.012% bromophenol blue; 8 mM EDTA) to 200 pmol of each oligonucleotide and heat at 95 °C for 2 min. Then, cool samples on ice and load on 15–20%, high-resolution polyacrylamide-7 M urea gels. Electrophoresis proceeds under denaturing conditions in 1X TBE buffer, 1.2 W/cm, in a darkroom.

2. The gel slices containing the fluorescently-labeled full-length primers are excised and soaked in 350 µL of elution buffer.

3. Incubate overnight at room temperature, in a darkroom.

4. Purify the DNA primers by phenol extraction, followed by chloroform:isoamilic alcohol extraction.

5. Extract the aqueous phase and precipitate the primers by the addition of 0.3 M sodium acetate, pH 6.0, and three volumes of absolute ethanol.

6. Pellet primer oligonucleotides as noted in steps 7–8 from Basic protocol 1.

7. Wash the DNA pellet by supplementing with 300 µL of 70% ethanol and proceed as indicated in steps 9–10 from Basic protocol 1.

8. Vacuum dry the samples and dissolve primers in 20 µL of RNase-free distilled water, by vigorous vortexing.

9. Measure DNA primers concentration by UV spectrophotometry (A_260_).

### 3.2. Primer Extension

1. Add 2.5 pmol of the NED-labelled primer to the (+) and (-) NMIA samples and mix by pipetting. Use 2.5 pmol of FAM- or VIC-labelled primer oligonucleotides for RNA sequencing ladders with 2 pmol of the target construct in separate tubes. An excess of primer may lead to a saturated signal in short-length products and the absence of full-length cDNA. A 1:1 RNA:oligonucleotide ratio is desirable.

2. Proceed to primer annealing by heating at 95 °C for 2 min and then snap cooling on ice for 15 min.

3. Prepare the RT reaction mix as indicated by the manufacturer and incubate the primer:RNA sample for 1 min at 52 °C. The sequencing reaction of each RNA sample using the same primer should be run in parallel. Sequencing of only one or two nucleotides could be enough. For that purpose, add 0.5 mM of the desired ddNTPs to each sequencing reaction. The choice of a specific ddNTP will depend on the specific sequence and the features of the RNA tested molecule. For IRES and 3′UTR of HCV, ddCTP, and ddTTP are good starting candidates.

4. Initiate primer extension by the addition of 1 µL of the SuperScript™ III enzyme mix and incubate samples at 52 °C for 20 min. Non-specific or premature reverse transcriptase stops by complex structural elements leads to an increase of non-specific signal in the untreated sample. The use of a heat-resistant reverse transcriptase is recommended to increase the temperature of the primer extension reaction. SuperScript™ IV enzyme is a good replacement to solve this problem. Premature signal decay and absence of full-length product may also be due to insufficient primer extension reaction time. Increase up to 1 h the reaction time.

5. Stop the reactions on ice.

6. Purify DNA samples using the BigDye XTerminator™ Purification kit (Applied Biosystems) and continue with the resolution of the cDNA products by capillary electrophoresis in an Applied Biosystems 3130xl Genetic Analyzer, as described [30]. The presence of the excess RNA template may interfere with the resolution of the capillary electrophoresis. Removing the RNA by treating the sample with 200 mM NaOH for 5 min at 95 °C prior to the electrophoresis may increase the resolution of the peaks.

## 4. Structural Analysis

Resolving cDNA samples by capillary electrophoresis using fluorophore-labeled primers has allowed the development of high-throughput techniques.

The extraction of reactivity data from the electropherograms is a challenging and, in many cases, time-consuming process. Different computational strategies can facilitate this task. One of the most useful tools is the QuShape software package [31]. It requires the use of two capillaries: the first one includes the treated reaction (+), along with one or two sequencing reactions of the target RNA; the second capillary is loaded with the non-treated reaction (-) and the same sequencing reactions. The use of sequencing reactions allows the proper alignment of the RT products. A complete and detailed procedure for the reactivity analysis using the QuShape software is reported in [31].

Finally, experimental constraints derived from biochemical and biophysical studies can be used to model RNA secondary structure. These include RNAFold [32] (http://rna.tbi.univie.ac.at/cgi-bin/RNAWebSuite/RNAfold.cgi, accessed on 21 September 2021) and the module ShapeKnots from RNAStructure [33] (https://rna.urmc.rochester.edu/RNAstructureWeb/, accessed on 2 September 2021). Practical user guides can be found on the corresponding websites.

RNA structure knowledge is a constantly growing field. This is reflected by the increasing number of publications reporting the development of new bioinformatics tools helping in the analysis of reactivity data and the interpretation of the results at both structural and functional levels. The RNAProbe web server [34,35] has been developed to simplify the entire routine by directly normalizing the reactivity output data from different probing assays. This new computational tool predicts the secondary structure of the RNA and yields high-resolution images and heatmaps, facilitating the interpretation of the results. Another tool, called RNAthor [36], has also been developed to facilitate the analysis and processing of statistical data.

In the last years, efforts have also been addressed in designing in silico tools able to predict SHAPE reactivity values for a given RNA sequence, which are then used to create 2D and 3D models. For example, the SHAPER web server can use both experimental and theoretical constraints to refine the folding and correlate it with other existing data [37]. Some tools have also incorporated phylogenetic and sequence co-evolution data [38] to generate libraries with the most favorable structural models.

These are just a few examples of how data derived from RNA probing techniques can be used to infer the folding of RNA molecules at nucleotide resolution, with high accuracy and reliability.

## 5. Summary and Future Perspectives

The development of new therapeutic strategies against emerging viral infections is, currently, of urgent need. During the last years, the potential of RNA molecules as antiviral drugs has been largely postulated and studied. These studies have allowed to overcome different challenges in the design of inhibitory RNAs, but have also evidenced the importance of the structure in the target region. Different in silico strategies have been designed to identify those regions in the RNA that are susceptible to being targeted by the inhibitors. However, these approaches show limited accuracy since they do not use experimental structure data. The improvements in RNA probing strategies and their combination with user-friendly in silico prediction tools have facilitated the tasks of designing RNA inhibitors, have significantly improved the accuracy of the results, and have also opened the door to the knowledge of RNA folding at a high-order level. In addition, the design of new drugs takes advantage of these experimental approaches to obtain data about optimal drug-binding pockets, including both dsRNA- and loop-binding compounds. New challenges are awaiting, such as deciphering the assembly pathways in viral RNA genomes, a critical step that is responsible for the final 3D RNA conformation, and therefore the function of the viral genome. Future innovations in the field of high-throughput RNA probing, including the synthesis of new chemical reagents and the development of more efficient RNA sequencing platforms, will generate a rich environment in which the analysis of RNA folding will be a requisite for many biomedical applications.

## Figures and Tables

**Figure 1 pharmaceuticals-14-01192-f001:**
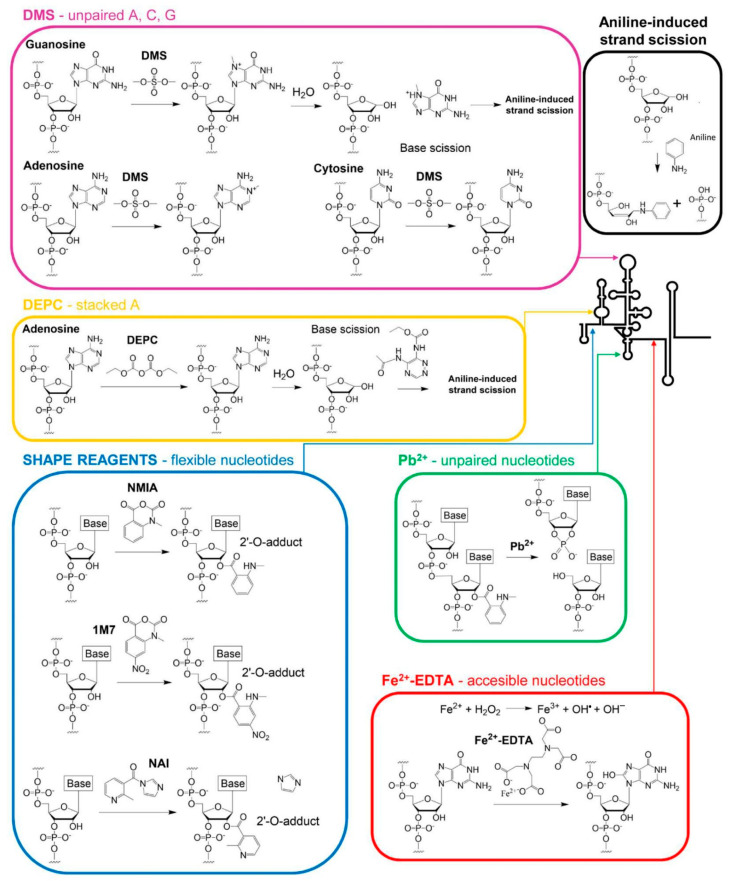
Schematic representation of the chemical reactions between an RNA molecule and the chemical reagents most commonly used for RNA structure probing. The figure shows the chemical structure of a specific chemical reagent and that of the nucleotides that react with it. The course of the reaction and the structure of the final products are also depicted. The conformational specificity of the reacting nucleotides of each reagent is represented by colored arrows in a diagram of the secondary structure of the 5′ end of the HCV RNA genome.

**Figure 2 pharmaceuticals-14-01192-f002:**
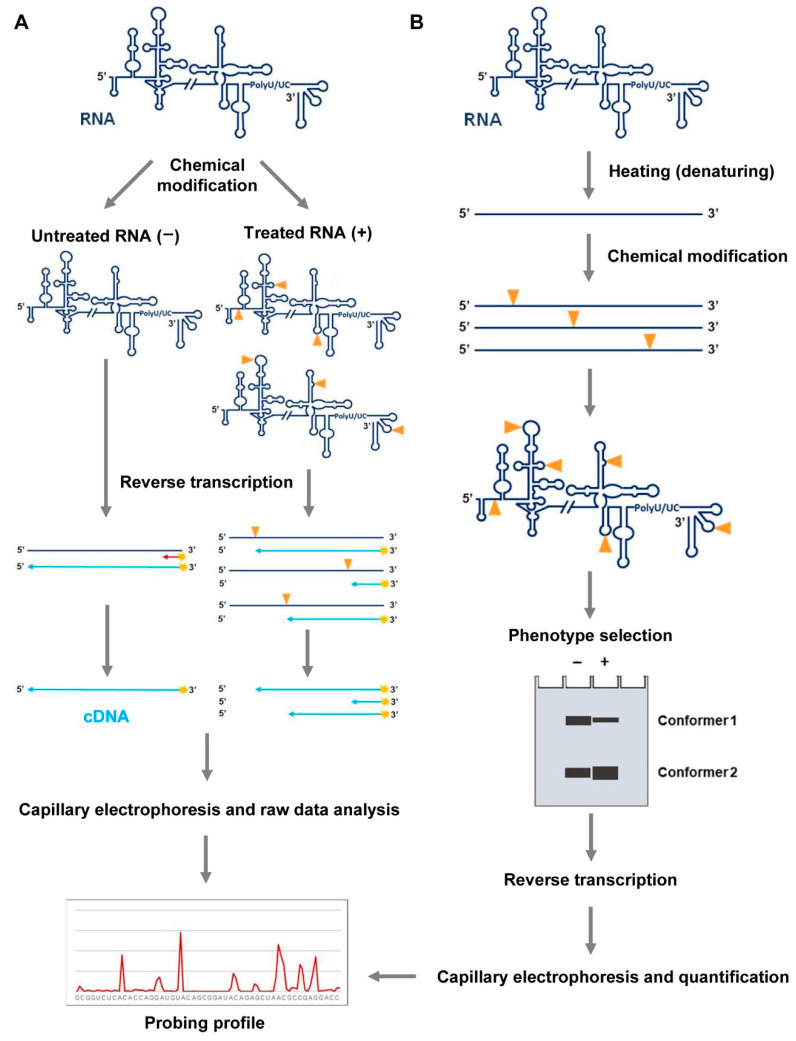
RNA probing. (**A**) RNA folding analysis by chemical probing or SHAPE analysis. The RNA is treated with chemical probes that covalently modify nucleotides at specific positions in a structure-dependent manner. Untreated samples must be also included in the assay for background normalization. These modifications, depicted by yellow arrows, act as stop signals in a reverse transcription (RT) reaction. Fluorescently color-coded labeled primers (in red) are used to map each modified residue. The resulting cDNA products are resolved by automated capillary electrophoresis. The raw data are scaled and normalized to obtain the relative reactivity values at each nucleotide, using the QuShape software. (**B**) Molecular interference strategy with SHAPE reagents (HMX). RNA molecules are modified with NMIA under denaturing conditions. The different conformers are partitioned by non-denaturing polyacrylamide gel electrophoresis. Modified positions, indicated as depicted in (**A**), are detected as stop signals in a reverse transcription reaction. The cDNA products are resolved by capillary electrophoresis and electropherograms are analyzed using the QuShape software. Data normalization yields the probing profile.

**Figure 3 pharmaceuticals-14-01192-f003:**
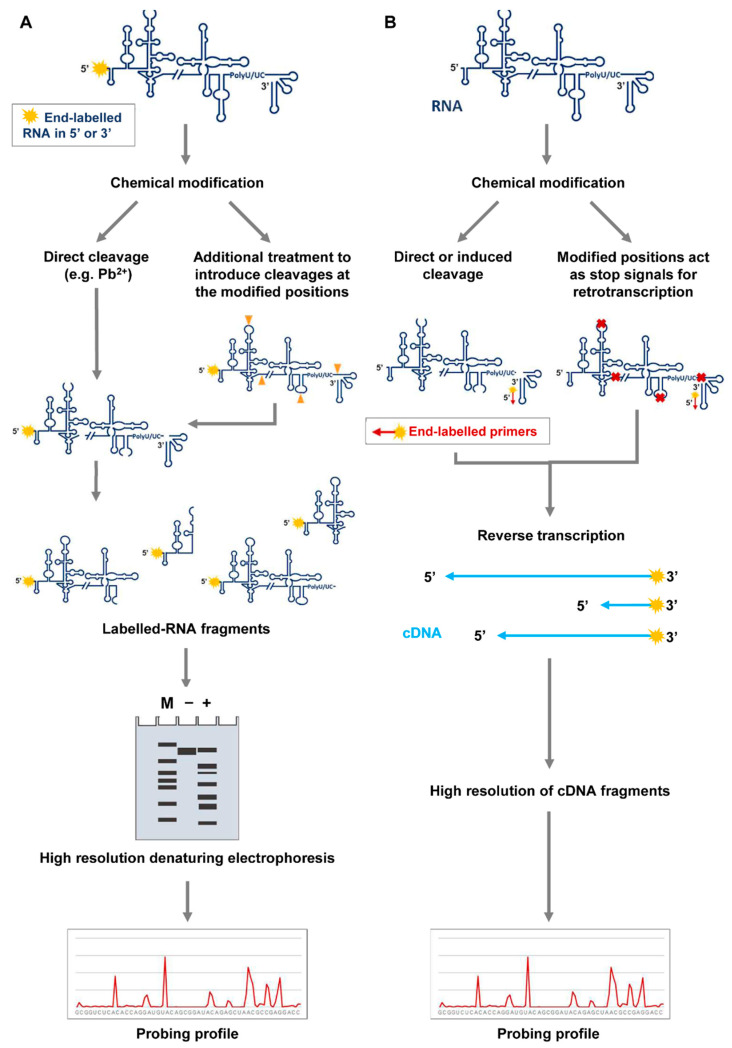
Detection of RNA modifications in probing assays. (**A**) RNAs can be labeled at either their 5′ or 3′ ends, then folded and subjected to modification. Chemical reagents like Pb^2+^ produce directly cleaved products, which can be resolved by high resolution denaturing polyacrylamide gels, along with molecular weight ladders for proper size assignment. Alternatively, some kind of modifications (denoted as in Figure 2A), like those induced by DMS on guanosine nucleotides, can be detected by inducing cleavage with additional treatments. Cleaved products are resolved as noted above. In both cases, the gels can be scanned and quantified by different bioinformatics tools to generate the probing profile. (**B**) Detection of modifications can be also accomplished by 5′ end-labeled oligonucleotides, which can be annealed at any position throughout the entire unlabeled RNA. Modified nucleotides are detected as stop signals in a reverse transcription reaction. The cDNA products are resolved by high-resolution electrophoresis or capillary electrophoresis, along with appropriate molecular sequencing reactions. Different software are available for cDNA quantification and the generation of the probing profile.

**Table 1 pharmaceuticals-14-01192-t001:** Common chemicals used for RNA probing.

RNA PROBING REAGENTS					
Application	Reagent	Target	Detection Methods	Applications	
				In Vivo	In Vitro
Base-specific reagents give information on the paired/unpaired state of nucleotides.	Dimethyl sulfate (DMS)	Unpaired A (N1), C (N3) and G (N7)	Primer extension. For G (N7), reduction of modified RNA and aniline-induced strand scission is previously required.	X	X
	2-keto-3-ethoxy-butyraldehyde (kethoxal)	Unpaired G (N1–N2)	Primer extension. Detection by RNase T1 hydrolysis can be used after modification for end-labeled RNA	–	X
	Diethylpyrocarbonate (DEPC)	A (N7), with a preference for stacked adenosines.	Aniline-induced strand scission and subsequent primer extension. Detection of cleavages on end-labeled RNA molecules is also possible, but the previous reduction of modified RNA and aniline cleavage is needed.	–	X
	1-cyclohexyl-3-(2-morpholinoethyl) carbodiimide metho-ptoluene sulfonate (CMCT)	Unpaired U (N3) and G (N1)	Primer extension	–	X
	Glyoxal	Unpaired G (N1–N2)	Primer extension	X	–
	Ethylnitrosourea (ENU)	Phosphate oxygen atoms.	Alkaline treatment and subsequent primer extension. Detection of cleavages on end-labeled RNA molecule is possible, but additional alkaline treatment is required too	–	X
Backbone-specific reagents give information on accessibility to solvent.	Ion Pb(II) (Pb^2+^)	Cleavage of the phosphodiester bond in unpaired nucleotides.	Primer extension with reverse transcriptase or detection of lead-induced cleavages on end-labeled RNA molecules	X	X
	1,10 -phenanthroline-copper(II) (OP-Cu)	Ribose backbone in regions accessible to solvent, with a preference for single-stranded regions.	Primer extension or detection of cleavages on end-labeled RNA molecules are possible	–	X
	Ethylenediamine tetraacetic acid-Fe(II) (Fe^2+^·EDTA)	Ribose backbone in regions accessible to solvent.	Detection of cleavages by RNA end-labeling or by primer extension	–	X

**Table 2 pharmaceuticals-14-01192-t002:** Frequently used SHAPE reagents for RNA structural mapping.

SHAPE REAGENTS					
Reagent Specificity	Reagent	Target	Detection Methods	Applications	
				In Vivo	In Vitro
Backbone-specific reagents informing about local nucleotide dynamics.	2-methylnicotinic acid imidazolide (NAI)	Ribose 2′OH group of residues in flexible regions. Effective differentiation of un- and paired adenosines with bias against guanosine and cytidine residues	Primer extension	X	X
	5-nitroisatoic anhydride (5NIA)	Ribose 2′OH group of residues in flexible regions. Over-reaction with adenosine.	Primer extension	X	X
	2-methyl-3-furoic acid imidazolide (FAI)	Ribose 2′OH group of residues in flexible regions	Primer extension	X	X
	N-methylisatoic anhydride(NMIA)	Ribose 2′OH group of nucleotides with slow dynamics	Primer extension	–	X
	1-methyl-6-nitroisatoic anhydride (1M6)	Ribose 2′OH group of nucleotides involved in stacking interactions	Primer extension	X	X
	1-methyl-7-nitroisatoic anhydride (1M7)	Ribose 2′OH group of unpaired nucleotides, with a preference for loops	Primer extension	X	X
	Benzoyl cyanide (BzCN)	Ribose 2′OH group of unpaired nucleotides, with a preference for loops	Primer extension	–	X
	N-propanone isatoic anhydride (NPIA) ^1^	Ribose 2′OH group of nucleotides with slow dynamics.	Primer extension	–	X

^1^ NPIA is a SHAPE Selection (SHAPES) reagent that is very similar to NMIA but it allows binding to a biotin molecule. Lately, cDNA/RNA hybrids are selected by streptavidin beads to eliminate the majority of background signals.

## Data Availability

Data sharing not applicable.

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
