# Peer review of "In Vitro Methods to Decipher the Structure of Viral RNA Genomes"

_pharmaceuticals, 2021, doi:10.3390/ph14111192_

Round 1

Reviewer 1 Report

In the review "In vitro methods to decipher the structure of viral RNA ge- 2
nomes" Romero-López et al. provide protocols for RNA structure probing using DMS, SHAPE and HMX. They also describe the analysis of the reaction products using the primer extension method with fluorescently labelled primers. All protocols are detailed and well described and will be of use for those new to the field.

Nonetheless, several issues should be adresses before publication of the review. The manuscript should be checked by a native english speaker as it contains several gramatical errors and very unusual choice of words. 
The authors mention 32P and fluoresencent end labelling as alternatives to primer extension analysis, but dont provide much information. It would be nice to point the reader to other reviews that describe these methods in more detail. Finally, part concerning the analysis of the probing data is basically limited to the citation of 3 programs or webservers. A more genereal treatment of the this topic (perhaps including potential pitfalls) would increase the usefulness of this review. 

Minor points:
table 1: ácido etilendiaminotetraacético (bottom) should be translated to english
page 6, line 178: 2-100 nM seems to be too low, are the authors sure about this?

Author Response

We thank the reviewer for his/her helpful suggestions and positive comments. We have addressed all the concerns, which have improved the quality of the manuscript.

The manuscript should be checked by a native English speaker as it contains several grammatical errors and very unusual choice of words.

The manuscript has been now reviewed by a person specialized in editing scientific texts.

The authors mention 32P and fluoresencent end labelling as alternatives to primer extension analysis, but don’t provide much information. It would be nice to point the reader to other reviews that describe these methods in more detail.

New references have been included.

A more general treatment of this topic (perhaps including potential pitfalls) would increase the usefulness of this review.

We agree with the referee in the interest of this point. However, we think that every bioinformatics tool is different and provides different information; this makes difficult in some way the writing of a general section. In this revised version we have included other recent tools that could be useful for the readers. We provide all the references in which these tools are described, including the detailed troubleshooting by the authors. These articles are very practical and complete guides.  

Minor points:

table 1: ácido etilendiaminotetraacético (bottom) should be translated to English

We thank the reviewer for this comment. We have corrected the table.

page 6, line 178: 2-100 nM seems to be too low, are the authors sure about this?

We have used this range of concentration in our experiments and it has worked properly in our hands. However, this concentration must be implemented for the total RNA amount and the length of the RNA of interest. It must be noted that these protocols are performed entirely in vitro

Reviewer 2 Report

The manuscript by Romero-Lopez et al. reviews the in vitro chemical probing methods of viral RNA secondary structures. The authors should make the below clarifications and corrections before the paper may be published.

Major comments:

  1. The authors may add schemes of a RNA chemical structure and show how nucleobases and sugars can react with the different chemical probes.
  2. Line 389, Discuss how the chemical probing methods can be used to identify drug binding sites on RNA, including dsRNA stem-binding ligands and loop-binding ligands.

Minor comments:

  1. Line 38-39, makes it tempting
  2. Line 43, against the same target
  3. Line 53, that take advantage of
  4. Line 61, sugar-phosphate backbone
  5. Lines 69-70, What do you mean by electronic dynamics? Is it conformational dynamics?
  6. Table 1 title: Common chemicals …
  7. Table 2 title: frequently used SHAPE reagents…
  8. Line 145 and other places: check the symbol for the degree Celsius
  9. Line 203, nucleobase may be changed to nucleotide
  10. Line 280, may be difficult for the analysis
  11. Line 381, …currently, of urgent need. During the last years…
  12. Line 387, experimental structure data.

Author Response

We thank the reviewer for the helpful comments. We agree with all the criticisms raised and we have modified the manuscript accordingly.

The authors may add schemes of a RNA chemical structure and show how nucleobases and sugars can react with the different chemical probes.

We have included an additional figure, named as Figure 1.

Line 389, Discuss how the chemical probing methods can be used to identify drug binding sites on RNA, including dsRNA stem-binding ligands and loop-binding ligands.

We have mentioned this idea as suggested by the reviewer.

Minor comments.

Line 38-39, makes it tempting; Line 43, against the same target; Line 53, that take advantage of; Line 61, sugar-phosphate backbone

These concerns have been corrected in the manuscript.

Lines 69-70, What do you mean by electronic dynamics? Is it conformational dynamics?

Electronic dynamics refer to the electronic movement in each reactive group of the target molecule. All the nucleotides in a flexible conformation are susceptible to react with SHAPE reagents, but the extent of such reactivity depends on the overall folding in which the nucleotides are embedded and the nature of the nucleotides that are in close proximity. This phenomenon has been described (see references 13-16 of the manuscript) and can be used to map specific conformations in the target RNA, such as helix stacking, loops or folding with different reagents. This is based in the timescales at which each SHAPE reagent can react, and there exists a clear correlation between timescale reactivity and the conformation of the target nucleotide. While NMIA preferentially reacts with residues that stabilize the architecture of the RNA molecule at small timescales (slow electronic dynamics), the second probe attacks “one-sided” stacked nucleotides, a typical conformation seen in bulges, turns, closing helix pairs, and long-range stacking interactions (fast electronic dynamics).

Table 1 title: Common chemicals …; Table 2 title: frequently used SHAPE reagents…; Line 145 and other places: check the symbol for the degree Celsius; Line 203, nucleobase may be changed to nucleotide; Line 280, may be difficult for the analysis; Line 381, …currently, of urgent need. During the last years…; Line 387, experimental structure data.

The manuscript has been modified according to reviewer suggestions.

Reviewer 3 Report

This “review article” is in fact, a protocol publication, which has its value for readers of Pharmaceuticals.

Using HCV RNA analysis (which previously published by the same group), the authors explain the principle of viral RNA secondary structure analysis for the purpose of designing accurate siRNA. The viral RNA sequences served as siRNA targets need to be expose to solution and not occupied by other part of viral RNA in secondary structure. Based on this concept, the authors describe step-by-step methods on how to use DMS, SHAPE, and HMX technique to modify the RNA, as well as how to identify the modification that facilitate the RNA structure analysis. This manuscript provides a useful detailed guidance for the researchers who are working in the similar field.

One would hope for a better description on general background of SHAPE technique. Sometimes the authors also explain certain concept in a manner that ignore earlier SHAPE development. For example, reading line 278-286 one would feel it is an issue being solved recently, whilst it has been successfully addressed as early as 2005, when the SHAPE technique was first developed. The reference list is also very limited. It would be a major problem for a literature review, but since it is essentially a method paper,  it probably serves the purpose.

Comments on the manuscript grammar

The manuscript requires additional proof-reading as there are grammatical errors throughout, particularly with regards to sentence structures. There are numerous sentences which either do not make sense due to the wording order or because words are missing words all together. Please see examples below:

Line 76:           ‘We have in vitro applied different probing…’

Changed to:    In vitro we have applied different probing…’

Line 95:           ‘Table 2. Most frequently SHAPE reagents used for RNA structural mapping’

Changed to:    ‘Table 2. Most frequently used SHAPE reagents for RNA structural mapping’

Line 30:           ‘copied and packaged to produce a new generation virions’.

Changed to:    ‘copied and packaged to produce a new generation of virions’.

Line 41-42:      ‘the siRNA technology has been extensively tested...’                     

Changed to:    ‘the siRNA technology which has been extensively tested...’              

There is also an over use of informal, or dated, words and phrases throughout the manuscript which detract from the scientific information being communicated.  For example, Line 53: ‘nowadays’, Line 140: ‘By the contrary’ and Line 150: ‘frail’. The use of the words ‘thus’ and ‘interesting’ are also scattered throughout the manuscript and should either be removed in the case of ‘thus’ or replaced with a more formal word(s) in the case of ‘interesting’.

Author Response

We thank the reviewer for his/her helpful suggestions. We have addressed all the criticisms raised.

One would hope for a better description on general background of SHAPE technique. Sometimes the authors also explain certain concept in a manner that ignore earlier SHAPE development. For example, reading line 278-286 one would feel it is an issue being solved recently, whilst it has been successfully addressed as early as 2005, when the SHAPE technique was first developed.

We agree with the reviewer that providing a more detailed view of the SHAPE methodology would be interesting. However, since this is a method manuscript, we have just included a general overview of the technique, paying attention to those concepts that are required to accomplish the technique. We have now added further details and we have re-written the above mentioned lines as suggested.

The reference list is also very limited.

We have included recent references.

Line 76: ‘We have in vitro applied different probing…’, changed to: In vitro we have applied different probing…’; Line 95: ‘Table 2. Most frequently SHAPE reagents used for RNA structural mapping’, changed to: ‘Table 2. Most frequently used SHAPE reagents for RNA structural mapping’; Line 30: ‘copied and packaged to produce a new generation virions’, changed to: ‘copied and packaged to produce a new generation of virions’; Line 41-42: ‘the siRNA technology has been extensively tested...’, changed to: ‘the siRNA technology which has been extensively tested...’            

Changes have been made in the text as suggested by the reviewer.
